# Pervasive male-biased expression throughout the germline-specific regions of the sea lamprey genome supports key roles in sex differentiation and spermatogenesis

Tamanna Yasmin[1], Phil Grayson [1,3], Margaret F. Docker[1] & Sara V. Good [1,2✉]

Sea lamprey undergo programmed genome rearrangement (PGR) in which ~20% of the genome is jettisoned from somatic cells during embryogenesis. Although the role of PGR in embryonic development has been studied, the role of the germline-specific region (GSR) in gonad development is unknown. We analysed RNA-sequence data from 28 sea lamprey gonads sampled across life-history stages, generated a genome-guided de novo super-Transcriptome with annotations, and identified germline-specific genes (GSGs). Overall, we identified 638 GSGs that are enriched for reproductive processes and exhibit 36x greater odds of being expressed in testes than ovaries. Next, while 55% of the GSGs have putative somatic paralogs, the somatic paralogs are not differentially expressed between sexes. Further, putative orthologs of some the male-biased GSGs have known functions in sex determination or differentiation in other vertebrates. We conclude that the GSR of sea lamprey plays an important role in testicular differentiation and potentially sex determination.

[1] Department of Biological Sciences, University of Manitoba, Winnipeg, Manitoba R3T 2N2, Canada. [2] Department of Biology, University of Winnipeg, Winnipeg, Manitoba R3B 2E9, Canada. [3] Present address: Department of Biomedical Informatics, Harvard Medical School, Boston, MA 02115, USA. ✉email: s.good@uwinnipeg.ca

The genetic structure and composition of germline and somatic cells typically remains constant throughout an organism's life span. However, under some conditions (e.g., cancer) and in some taxa, the genetic composition of cells varies by type and/or developmental stage[1,2]. Included in this is the unusual process of programmed genome rearrangement (PGR), in which either portions of chromosomes (chromosomal diminution) or entire chromosomes (chromosomal elimination) are removed during embryonic development, thereby reducing the genomic content of descendent cells by up to 90%[3]. Although the frequency of PGR across metazoans is unknown, it has been observed in more than 100 vertebrate and invertebrate species from nine major taxonomic groups[3], including lampreys[4–7]. In sea lamprey (*Petromyzon marinus*), flow cytometric measurements of DNA content in the germline (testes) vs. somatic (blood) cells indicate that ~20% (~500 Mb) of the germline genome is eliminated during PGR[5]. Further studies have shown that PGR in sea lamprey, which occurs ~3 days post-fertilization (dpf), shares conserved features with PGR in other agnathans[4–7]. PGR entails the removal of both repetitive and single-copy sequences, and the targeted chromosomal regions are bundled into transcriptionally inactive heterochromatin and jettisoned from all but the primordial germ cells[6,8].

Many hypotheses have been posited regarding the biological significance of PGR, including gene silencing, dosage compensation, position effects on gene expression, germline development, and sex determination[1,9–11]. Most organisms that undergo PGR, such as ciliates, sciarid flies (*Sciara coprophila*), songbirds, hagfishes, and lampreys, exhibit high levels of histone methylation on the chromosomes or chromosomal regions eliminated during PGR[12]. In sciarid flies, the elimination of one or two paternally inherited X chromosomes in the pre-somatic cells determine sex[9]. In the zebra finch (*Taeniopygia guttata*), chromosomal diminution of a germline-restricted chromosome (GRC) occurs during early embryonic development; the genes in the GRC have higher expression in the ovary than the testis, and the GRC is later eliminated from mature sperm and thus transmitted only through the oocytes[13–15]. In sea lamprey, gene ontology analysis indicates that the GSR is enriched for genes involved in development and germline maintenance[4]; thus, it has been argued that PGR permits the expression of genes that are beneficial during the early stages of embryonic development but that could be harmful if misexpressed in somatic tissues at later developmental stages[5,6]. However, the possible role of the GSR in gonadogenesis is unexplored.

In sexually reproducing taxa without PGR, primordial germ cells (PGCs) are set aside shortly after fertilization to protect the germ cells from mitotic damage; the cells to be set aside are determined via inheritance, induction, or a combination of the two[16]. Once defined, PGCs exhibit tightly coordinated gene expression that leads to subsequent germ cell development and differentiation in both sexes at the appropriate developmental time. In lampreys, however, the germline cells are those present at fertilization, and somatic cells originate ~3 dpf, when PGR is initiated[5]. This intriguing reversal of events is heightened by the ongoing enigma of their sex determination. Lampreys do not have heteromorphic sex chromosomes, and there is no evidence to date of genomic differences between males and females; sex may be determined by genetic factors in the germline genome, environmental factors, or a combination of the two[17,18].

Here, we used RNA-sequence (RNA-seq) data from 28 sea lamprey gonads sampled at different life-history stages and in both sexes to generate a gonadal superTranscriptome, and we examined the function, expression, and evolutionary relationships of sex-biased genes, particularly in the GSR. We identified 638 genes in the GSR, which we will refer to as germline-specific genes (GSGs). Many of these GSGs have germline-specific paralogs, such that the 638 GSGs belonged to 163 gene families. These 638 GSGs were, overall, highly expressed during spermatogenesis, but lowly expressed during oogenesis and in larvae with histologically undifferentiated gonads. The observation that the genes in the GSR appear to be present, but expressed at low levels, in undifferentiated larvae and females suggests that the male-specific expression is due to regulatory changes, as opposed to there being a male-specific germline sequence. Further, we found that ~55% of the GSGs also have paralogous copies in the somatic genome that overall do not exhibit sex-biased expression and ~19% have putative orthologs in other taxa, including homologs of genes involved in sex determination and spermatogenesis. Using publicly available RNA-seq data from embryos 1–5 dpf, we confirm that the GSGs expressed during sea lamprey gonadogenesis are either not expressed or are lowly expressed during early embryo formation. Collectively, these results suggest that a major role of the GSR is in testicular differentiation and potentially sex determination. PGR in sea lamprey may serve to reduce conflict of genes under sexual selection, a hypothesis further supported by the highly duplicated nature of genes in the GSR and their association in sexual differentiation and determination pathways in other taxa[3].

## Results and Discussion

**GSGs show predominantly male-biased expression and have a key role in gametogenesis.** We used RNA-seq data from 28 sea lamprey gonads sampled across a range of developmental stages to generate a gonadal superTranscriptome using the Necklace pipeline[19]. Stages included larvae with undifferentiated gonads, female larvae following the onset of oogenesis, sexually mature (adult) females, prospective male larvae (i.e., those in which the gonad was still histologically undifferentiated but which were beyond the size at which ovarian differentiation is complete), males undergoing testicular differentiation following the onset of metamorphosis, and sexually mature (adult) males (see Supplementary Fig. 1 and Supplementary Table 1). This revealed a large number of genes that were highly expressed during male but not female gonad development; these genes were physically linked and mapped to chromosome 81 and to many unplaced scaffolds based on the Vertebrate Genome Project (VGP) reference assembly. Thus, we sought to define which of the genes in our gonadal superTranscriptome mapped to the GSR.

The GSR in sea lamprey was previously identified for an earlier release of the sea lamprey genome (www.stowers.org) by mapping sperm (germline) and blood (somatic) DNA from a single male back to the germline assembly, and applying the DifCover algorithm[20]. Thus, we used a modified version of the DifCover pipeline used for that analysis[21] to define the coordinates of GSR in the VGP assembly. Accordingly, GSRs were designated as regions in which the read coverage of sperm DNA was >2-fold more than the read coverage of blood DNA. Based on the VGP reference genome, a total of 5253 genomic intervals were mapped by DifCover, of which 919 segments had an enrichment score (log2(standardized sperm coverage/blood coverage) >2). The total span of the GSR-inferred regions consisted of more than 27 Mb (Supplementary Data 1, Supplementary Fig. 2).

We then used the segment enrichment scores to assign genes from our gonadal superTranscriptome to either the GSR or somatic genomes. Using an earlier scaffold-based assembly of the sea lamprey germline genome (available at SIMRbase), Smith et al. (2018) identified ~13 Mb including 356 protein-coding genes in the GSR[20]. On the other hand, using the VGP assembly which consists of 85 chromosomes and 1195 unassembled scaffolds, we assigned the entirety of chromosome 81, as well as

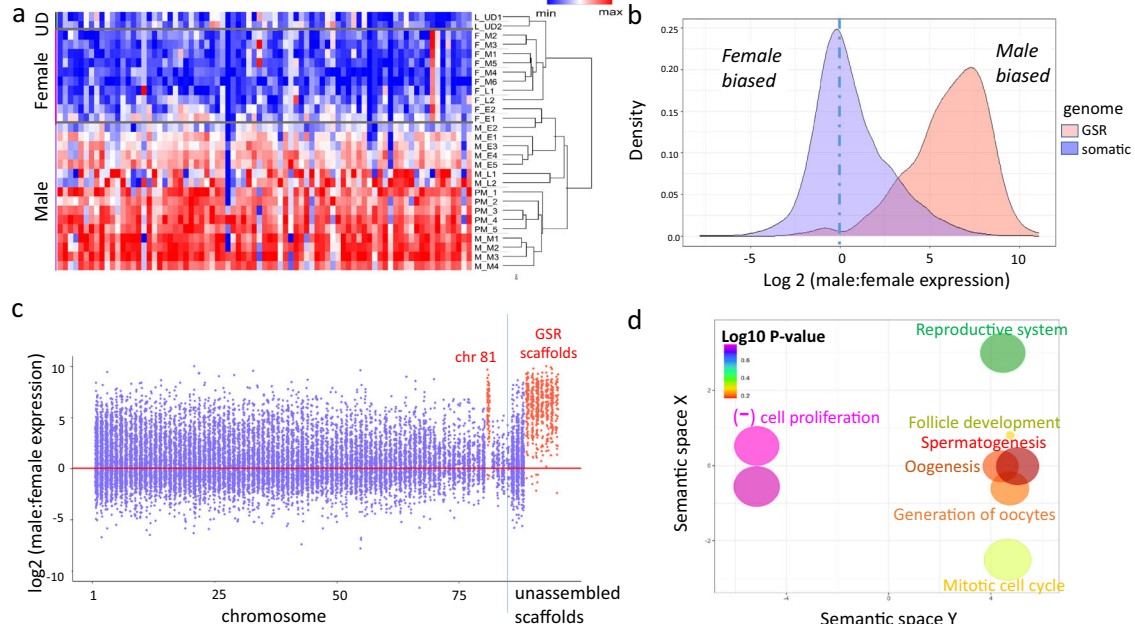

**Fig. 1 Identification of genomic location of the germline-specific genes (GSGs) in the VGP genome and their expression in the sea lamprey gonadal samples used in this study. a** Heatmap showing GSG expression pattern of all samples used in the study; UD stands for undifferentiated larvae (see Supplementary Fig. 3 for full heatmap). **b** Density plot of the log2(male:female) ratio of normalized gene expression. When x = 0, average expression across all females = males. For genes in the somatic genome, the density peaks at ~x = 0 but is right skewed, while for genes in the GSR, the density peaks at ~x = 7.5, showing that genes in the GSR are male-biased. **c** Scatterplot showing the log2(male:female) normalized gene expression across all chromosomes and concatenated scaffolds in the VGP assembly of the sea lamprey genome; regions identified as belonging to the GSR are colored in red, while those in the somatic genome are colored in blue. **d** Gene ontology (GO) term enrichment analysis of the GSGs where colors indicate the log10 of the false discovery rate-corrected P-value (PANTHER overrepresentation test, with a Fisher exact test for significance and filtering using a false discovery rate of 0.05); circle size denotes fold enrichment above expected values.

177 scaffolds (Supplementary Fig. 3) to the GSR, while the remaining 84 chromosomes and 1018 scaffolds were not germline-enriched, suggesting that they are found in the somatic genome. In total, 638 genes from our gonadal superTranscriptome mapped to the GSR; these 638 genes corresponded to only 163 unique gene family names based on our combined Trinotate and reference genome annotation pipeline (Supplementary Data 2), with approximately half of the GSGs occurring in a single copy but the other half occurring in 2–77 duplicated gene copies (Supplementary Data 3, Supplementary Fig. 4). Importantly, we did not identify any germline-specific regions on scaffolds or chromosomes that also contain regions that appeared to be present in the somatic genome. This supports previous work that determined that PGR in lampreys is more likely to involve chromosome elimination than diminution[8].

The expression analysis of the GSGs revealed that out of the 638 GSGs all were expressed in one or more stages of male gonad development, but expression in female gonads and undifferentiated larvae was predominantly low (Fig. 1a, Supplementary Data 2). Across both prospective males and definitive males (see Supplementary Fig. 1 for stage description), the median normalized gene count of GSGs was 208, and 31 genes had a median count of <5, while across the 9 differentiated ovarian samples and in the undifferentiated larvae, the median expression was 10.9 and 49.2 respectively, but most genes exhibited no or low expression (445 and 483 genes respectively had a median count <5) (see Fig. 1a and Supplementary Fig. 5 for full heatmap, Supplementary Data 2 for gene counts). To probe this sex-biased expression genome-wide, we calculated the average log2 male:female expression of all genes in definitive males (early, mid, and late developmental stages) and prospective males relative to definitive females. This revealed the surprising result that almost

all of the genes in the GSR exhibited male-biased expression during gonad development, while genes in the somatic genome were, overall, equally likely to be expressed in the female or male gonad (Figs. 1b, c, Supplementary Fig. 6). This is most clearly seen by comparing the density of log2 male:female gene expression ratio: the density peaked at x = 0 for genes in the somatic genome, but at x = ~7.5 for genes in the GSR (Fig. 1b).

A possible explanation for this observation could be that females do not have the same GSR as males, since the reference genome for sea lamprey was generated using sperm DNA. However, this does not appear to be the case; genes expressed in the developing female gonad were found on chromosome 81, as well as on many of the unassembled scaffolds (Supplementary Data 2, Supplementary Fig. 7). To examine this more closely since RNA-seq reads may map to spurious locations, we aligned individual BAM files from both male and female gonad samples to the indexed superTranscriptome and annotation files using the Integrative Genome Viewer (IGV)[22]. This revealed that for the few genes expressed in both testes and ovaries, the reads aligned to overlapping genomic (exons) structures (e.g., see Supplementary Fig. 8a, 8b). This suggests that females harbor the same GSR as males, but that it may be silenced via epigenetic controls such as DNA hypermethylation or histone modifications.

Next, we analyzed the functional enrichment of the Gene Ontology (GO) terms associated with the GSGs and identified that they were associated with 26 pathways, of which *wnt* signaling and *E-Cadherin* signaling pathways each represented 16.4% of the total hits (Supplementary Fig. 9). Other critical pathways include the insulin/insulin growth factor (*igf*) pathway, gonadotropin-releasing hormone receptor (*gnrhr*) pathway, transforming growth factor-beta (*tgfb*) signaling pathway, and fibroblast growth factors (*fgf*) pathway, which contained 2.7, 2.7,

2.7, and 1.4% of all hits, respectively (Supplementary Fig. 9). We performed a Fisher exact test to identify which GO terms were overrepresented based on the proposed molecular function of the GSGs (Supplementary Data 4). The molecular functions with the lowest FDR *P*-values were reproductive system development, positive and negative regulation of cell population proliferation, ovarian follicle development, oogenesis, and spermatogenesis. Collectively, this demonstrates that the functional ontology of GSGs is enriched for GO terms related to reproductive developmental processes (Fig. 1d, Supplementary Data 5).

The majority of investigations into the function of PGR in vertebrates has found that it is associated with the elimination of sex chromosomes, and thus it is argued that PGR can be an extension of dosage compensation in which epigenetic inactivation of genes on sex chromosomes is used to equalize gene expression in males and females[12,23]. In lampreys, the majority of eliminated segments belong to 12 small germline-specific chromosomes[21]. It is not clear if one or more of the eliminated chromosomes could serve as a sex chromosome, but fluorescence in situ hybridization with germline-specific probes revealed a single karyotype in sea lamprey embryos that were actively undergoing chromosomal elimination (1.5–2 dpf), arguing against sex-specific differences[21]. Given the important role of the GSR in male gonadogenesis identified here, a complete sequencing and assembly of the GSR in sea lamprey female gonad is warranted, so that it can be compared to the male germline genome in detail.

The mechanism of sex determination in lampreys remains unknown, and may involve both genetic and environmental factors[17,24–26]. The single elongated gonad remains histologically undifferentiated for up to several years, and the differentiation process is asynchronous in females and males[17]. Ovarian differentiation occurs in the larval stage, following synchronized and extensive meiosis and oocyte growth. A few small oocytes may also appear in future males, but testicular differentiation does not occur until the onset of metamorphosis ~2–3 years later, when resumption of mitosis in the remaining undifferentiated germ cells produces spermatogonia[27]. It also appears that some larvae may be capable of undergoing sex reversal to males following initial ovarian differentiation[26]. Thus, a suite of genes could be turned on to initiate testicular differentiation. Our data suggests that female sea lamprey gonads harbor the same GSR as males but, with the exception of some rRNA and ribosomal protein-coding genes, females exhibited low expression of the GSGs (Supplementary Fig. 7). Male-biased sex ratios under conditions of high larval density or slow growth have led to suggestions that primary sex differentiation in lampreys is influenced by environmental factors[24,25]. Putative genetic females of European sea bass (*Dicentrarchus labrax*) were masculinized after exposure to high temperatures, which was accompanied by changes in DNA methylation of the *cyp19a* (essential for ovarian development in mammalian and non-mammalian vertebrates) promoter[28]. This provides a specific mechanism for how environmental factors such as temperature and density could influence the activation or silencing of genes via epigenetic modification in the GSR and, at least partially, control sex determination in sea lamprey.

**Somatic paralogs of GSGs are expressed differently than germline paralogs**. We observed that many of the GSGs had duplicated copies: of the 163 GSGs, 92 were found to have one or more paralogous copies in the GSR (Supplementary Data 3), while 89 had putative paralogs in the somatic genome, suggesting that some of the GSGs may have been recruited to the GSR to play specific roles in gametogenesis. The somatic paralogs of the

GSGs were found distributed throughout the entire somatic genome, on every chromosome except chromosome 49 (Fig. 2a, Supplementary Data 6). To assess whether the somatic paralogs of the GSGs exhibit similar sex-biased expression, we selected one paralogous gene per genome (somatic and germline) and generated a heatmap to compare somatic vs. GSR expression of the paralogous genes (Fig. 2b). In keeping with the somatic-wide pattern (Fig. 1a, this demonstrated that the somatic paralogs of the GSGs do not exhibit the same sex-biased expression (Fig. 2b).

*Comparison of male-biased gene expression across stages and genomes*. Given the evidence of male-biased gene expression in the GSR, we next examined whether the GSGs had uniform expression across male gonadal developmental stages relative to male-biased genes in the somatic genome. For this analysis, we applied a stricter criterion to identify male-biased genes (see Methods), and selected the top 20% of genes showing a $\log_2FC > 2$ in males:females. In total, we identified 1270 strongly male-biased genes across the sea lamprey genome (of 18,945 total genes), of which 409 (of 638 total genes) were found in the GSR and 861 (of 18,307 total genes) were found in the somatic genome, indicating that genes in the GSR have a 36× higher odds of exhibiting strong male-biased expression (OR = 36.5068 where $P < 0.0001$).

Using the normalized counts of transcripts exhibiting strong male-biased expression, we compared the proportion of total transcripts in early, mid, and late testicular development and in prospective males across genomes (somatic vs GSR) using a repeated measures mixed model design in which gene nested in the genome was a random effect, and the stage was a repeated measure (Supplementary Data 7, Supplementary Fig. 10). This showed that there was a higher proportion of genes expressed in males in mid-testicular development and in prospective males in the GSR compared to somatic genomes, and a significantly lower proportion of genes expressed in early and late testicular development in the GSR relative to the somatic genome (Fig. 2c, Supplementary Data 7 and 8).

To visualize the stage-specific bias in gene expression of GSGs, we plotted the relative proportion of transcripts expressed in each of the three male gonadal stages (early, mid and late), as well as in prospective males and the pooled sum of transcripts expressed at any female stage (Fig. 2d, Supplementary Data 9). This underscores that there is a similar pattern of expression across all genes in the GSR: high gene expression in prospective and mid gonadal stage males, but zero to very low expression in females. These findings are similar but distinct from those in zebra finch: the chromosomes undergoing chromosomal diminution and the genes eliminated during PGR in zebra finch do not exhibit strong sex-biased expression; however, there was a significant enrichment for genes involved in ovarian development[14]. On the other hand, in a sciarid fly, the elimination of one or two paternal X chromosomes in all somatic cells determines the sex of the embryo[9]. Small intestinal parasites (*Strongyloides ratti* and *Strongyloides papillosus*) also use this elimination event for sex determination, where either one whole chromosome or a part of the chromosome becomes lost in males, but not in females[29,30].

Here, we find that the GSGs show comparatively higher expression in both presumptive males and males in mid-testicular development (Supplementary Fig. 1). In mid-developmental males, the germ cells are undergoing mitotic proliferation and producing spermatogonial Type B cells (see Supplementary Fig. 1). In prospective males, the gonads are histologically undifferentiated (and may remain so for another 1–3 years[17]), while females from the same size class have oocytes arrested in Meiosis I. The finding of high expression of GSGs in undifferentiated prospective males but not in females of the

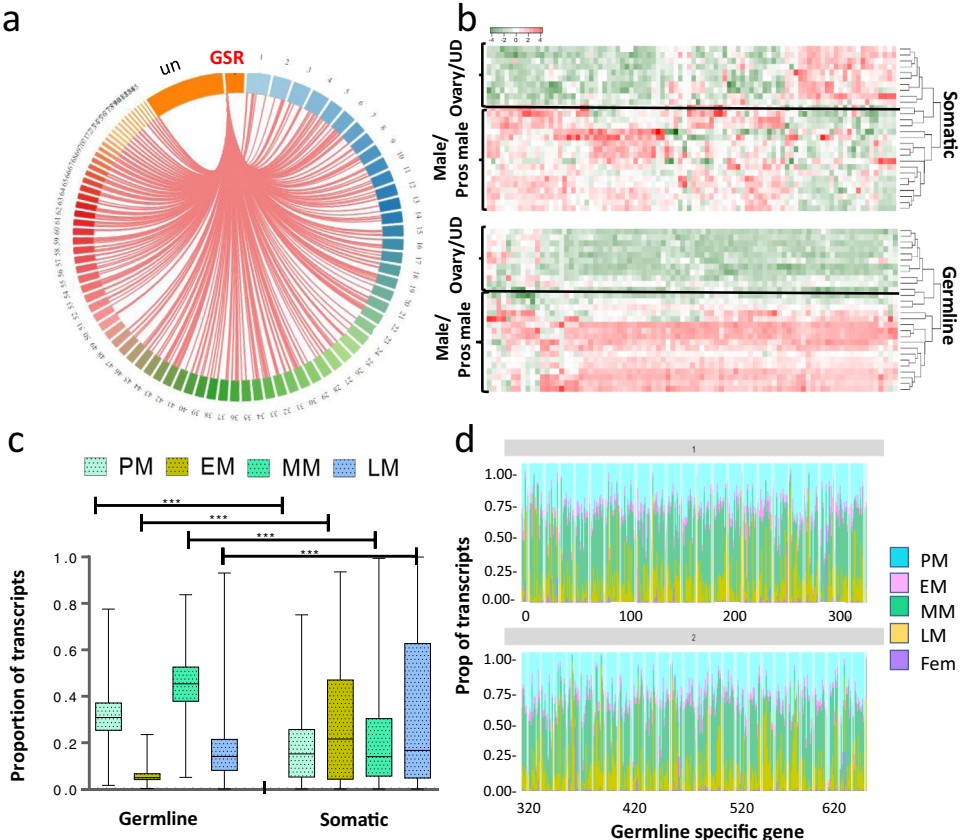

**Fig. 2 Somatic paralogs of GSGs are expressed differently than germline paralogs. a** Circos plot indicating the link between genes in the GSR with putative somatic paralogs in the sea lamprey genome. Chromosome 81 and enriched scaffolds are indicated as GSR and non-enriched scaffolds are indicated as un (unplaced scaffolds in somatic genome). **b** Heatmap showing the relative expression of genes that have paralogs in both the GSR and somatic genomes in males and prospective males (pros male), females (ovary), and undifferentiated larvae (UD). **c** Box plot showing the gene expression differences of somatic and GSR paralogs of GSGs in prospective males (PM), and early, mid, and late males (EM, MM, and LM, respectively). **d** Proportion of the transcripts for each of the 638 GSGs expressed in prospective, early, mid, and late males (PM, EM, MM, LM) and females (Fem).

same size class suggests that the GSR plays an important role in sex differentiation and potentially sex determination in sea lamprey, with high expression of GSGs leading to testicular differentiation and development in males and gene silencing resulting in ovarian differentiation in females.

Our analysis suggests that the GSR is present in ovaries, but is expressed at low levels (Supplementary Fig. 8a, b, Supplementary Data 2). As noted, one possibility could be that differential DNA methylation of the developing male or female gonad is involved in sex determination or differentiation. DNA methylation is a common process of epigenetic modification with known roles in gene regulation, embryogenesis and increasingly, sex determination[31] which has, interestingly, become a more important form of gene regulatory control throughout deuterostome evolution[32]. On the other hand, Histone 3 Lysine 4 (H3K4) methylation is abundant at active promoters, whereas H3K27 methylation is associated with transcriptional inhibition[33,34]. At the bipotential stage in primordial germ cells, sex-determining genes may be poised for either activation or repression, with activation requiring the removal of H3K27me3 repressive marks and repression requiring the maintenance of existing H3K27me3 repressive marks[35]. In lampreys, the chromosomal regions that are eliminated during PGR experience epigenetic silencing via DNA and histone methylation[21]; therefore, it is feasible that comparable epigenetic silencing via DNA and histone methylation may play a role in sex determination as well.

An increasing number of studies indicate roles for both chromatin modification and changes in DNA methylation in sex determination. Proteins involved in histone modification or the opening of dense chromatin play a key role in germ cell fate and gonad development in mice[36,37]. On the other hand, changes in DNA methylation of the promoter region of *Sry*, essential for mammalian testis development, regulate the temporal expression of *Sry* in mice[38]. A recent study in zebrafish (*Danio rario*) found that DNA methylation plays important functions in germline development as well as in sexual plasticity[39]. Given the clear role for the GSR in male spermatogenesis, we wanted to probe the expression of the GSR during early development bracketing PGR itself. To this end, we analyzed publicly available RNA-seq data from sea lamprey embryos that span the PGR (1–5 dpf). Of the 638 genes we identified in the GSR, only 186 were expressed during early embryogenesis. Of these 186 genes, 146 were expressed prior to PGR and 111 were expressed post-PGR. However, only 20 had an average gene count >50 post-PGR and 18 pre-PGR (Supplementary Data 10), while the five most abundantly expressed genes code for ribosomal proteins. We then compared the expression of the 186 GSGs expressed in pre- and post-PGR embryos with our male and female gonad samples, and found that the GSGs exhibited very low expression in females and embryos, but high expression in male gonads (Supplementary Fig. 11). This further supports the hypothesis that the role of the GSR in sea lamprey is predominantly to support male gonadal development.

**Phylogenetic relationship of GSGs provides evidence of diversified genes involved in sex-determination and differentiation.** Lampreys diverged from the jawed vertebrate lineage more than 500 million years ago[40,41], either after the two rounds (2 R) of whole genome duplication (WGD) that occurred in early vertebrate evolution[42,43], or more likely after 1 R[20,44,45]. A recent study suggested that, after the 1R tetraploidization, lampreys underwent an additional hexaploidization[46]. Since lampreys have an unusual vertebrate ploidy state, it proved impossible to perform a reliable test of positive selection at the amino acid level (which requires essentially gapless alignments) for the germline genes in agnathans (lampreys and hagfishes) relative to other vertebrates. Thus, we selected a few genes which have important roles in gametogenesis in other species for phylogenetic analyses.

Gene trees were reconstructed using the output from OrthoFinder and the orthologs of sea lamprey GSGs in 10 other chordates (see Supplementary Fig. 12), and they were combined with our data on gene annotations and genomic location (GSR vs. somatic) in sea lamprey. This revealed that the *cadh* gene family was highly duplicated in both the germline and somatic genomes of sea lamprey (16 vs. 15 duplicates, respectively) and hagfish (Supplementary Data 11). In particular, *cadh2* had undergone a divergent expansion in the GSR in sea lamprey (Supplementary Fig. 13a); agnathans had witnessed an expansion of a somatic cluster of genes related to vertebrate *cadh1/cadh3/cadh13* as well as an expansion in both the somatic and germline genomes of a novel cadh paralog (bottom of Supplementary Fig. 13a). Phylogenetic trees for *hykk* (Supplementary Fig. 13b), *sycp1* (Supplementary Fig. 13c), and *adgrl* (Supplementary Fig. 13d) depicted similar patterns of one or more highly duplicated germline lineages that were sometimes interspersed with closely related somatic paralogs (*hykk* and *scyp1*). Overall, however, they showed clades of highly diversified germline lineages marked by long internal branch lengths, indicating that the GSGs exhibit independent evolution for variable periods of time and may be subject to positive selection.

We searched the literature for evidence that any of the 163 unique gene names we identified in the GSR are associated with sex determination and/or differentiation in other taxa (Supplementary Data 1). Some of the GSGs had been found to exhibit female-biased expression in later-diverging vertebrates, and some were involved in ovarian development, suggesting that the tissue of expression (gonad) may be conserved, but the function (male vs. female gonadogenesis) is not. Importantly, however, we found orthologs or paralogs of most of the core genes involved in sex determination across vertebrates, for example, fibroblast growth factor 8 (*fgf8*), which is involved in sex determination in mice[47–49], as well as fibroblast growth factor receptor 3 (*fgfr3*), which is involved in sex determination in sturgeon (*Acipenser dabryanus*)[50]. Further, we identified a novel *fgfr3*-like gene in the germline genome, which could be a receptor for *fgf8b*, which was also located in the germline genome. The *fgfr3*-like gene was not identified by OrthoFinder as an ortholog of the somatic copy of *fgfr3*; thus, we downloaded the canonical coding sequences for *fgfr3*, and a related gene also present in the somatic genome, *fgfrl1*, from eight post-2R taxa and reconstructed a ML tree with bootstrap support (Fig. 3a). This revealed that the germline sequence of the *fgfr*-like coding sequence is more closely related to *fgfr3* in the sea lamprey somatic genome and to the *fgfr3* in later-diverging vertebrates (bootstrap support 100%), while the somatic copy of *fgfrl1* groups with the *fgfrl1* sequences from later vertebrates and there is no paralog in the GSR (100% bootstrap support). Examination of the expression of these three genes, as well as the possible receptor for the germline gene *fgf8b*, indicates that the germline copy of *fgfr3* and *fgf8b* had very low expression in female gonads, and somewhat higher expression in male

gonads: notably, *fgf8b* was most highly expressed in prospective male and mid-stage male gonads (Fig. 3b–e). Given their role in sex determination in other vertebrates, sea lamprey germline genes *fgf8b* and *fgfr3* warrant further investigation as possible loci involved in sex determination.

**Evolutionary conservation of GSGs and their function in vertebrate sex determination/differentiation and spermatogenesis pathway.** Genes in the GSR are expected to be released from the dosage sensitivity constraints of genes in the somatic genome[4] and may show conservation of gene functions related to gonadal sex determination and differentiation in other vertebrates. We thus hypothesized that genes in the GSR: 1) do not originate from a single linkage group in the pre-vertebrate ancestor, and do not map to a single linkage group in the post-2R vertebrate genome, 2) exhibit accelerated evolution either via high rates of duplication and/or amino acid change, and 3) have known roles in sex determination or spermatogenesis in other vertebrates. To this end, we performed comparative mapping of genes in the sea lamprey GSR to an earlier chordate (*Branchiostoma belcheri*) and to nine post-2R taxa. Of the 163 unique gene names identified in the GSR, orthologs with variable levels of conservation across chordates were identified for 31 genes (Supplementary Data 11). Some of these genes were found predominantly as a single copy in most taxa, whereas in sea lamprey, we found a single copy in the GSR but multiple paralogs in the somatic genome (*rpab4, rlp37A, mid2bp,* and *hsop3*), multiple paralogs in both the GSR and somatic genomes (*scyp1*) or a single copy in the GSR and somatic genomes (*fgfr3*), or a single copy in the GSR but no copy in the somatic genome (e.g., *agrl3, cxb1, hsop3, rpab4*) (Supplementary Fig. 12, Supplementary Data 3).

On the other hand, some of the genes in the GSR were found in multiple paralogs in later vertebrate genomes, and in multiple copies in the GSR and/or somatic genomes in sea lamprey (*agrl3, cxb1, spop1, lpar1, cadh2, lrrn1, mlcl1*) (Supplementary Fig. 12). Of the 31 genes assigned to an orthogroup, 23 were also identified in *Branchiostoma* (Supplementary Data 11, Supplementary Fig. 12), and 9 were present only in the GSR (not the somatic genome), suggesting that some of the lamprey GSGs were not novel. This suggests that the genes involved in spermatogenesis in the GSR were independently duplicated into the GSR. Comparison of the genes in the sea lamprey GSR with those in other agnathans will be needed to understand the evolutionary and selective forces shaping the number and rate of evolution of genes in the GSR.

In addition to *fgf8b/fgfr3*, we identified other core genes involved in sex determination or early sexual differentiation in later-diverging vertebrates such as R-spondin (*rspo1*), beta catenin 1 (*ctnnb1*), and 8 copies of wnt paralogs (*wnt5a, wnt5b, wnt7a, wnt7b*), which are key genes for the Wnt pathway which initiates testicular differentiation (Supplementary Table 2)[51,52]. Further, several of the gene families known to be essential for spermatogenesis were highly duplicated. For example, cadherins (*cadh*) are responsible for maintaining the integrity of testis structure[53], *scyp1* is important for early meiotic recombination during spermatogenesis[54], cyclins (*ccnb*) are essential for cell progression during distinct phases of the male spermatogenesis pathway[55], RNA binding proteins (*rbm*) play diverse and important roles in spermatogenesis including testis-specific splicing[56], and the absence of *rbm46* (present in 16 copies in the sea lamprey GSR) is associated with male infertility in mice[57]. Other important genes (e.g., *sox9* and *cbx2*, which play roles in stabilizing the male differentiation pathway) are present in the somatic genome of sea lamprey. Intriguingly, we found that these genes were highly expressed in the gonads of prospective males

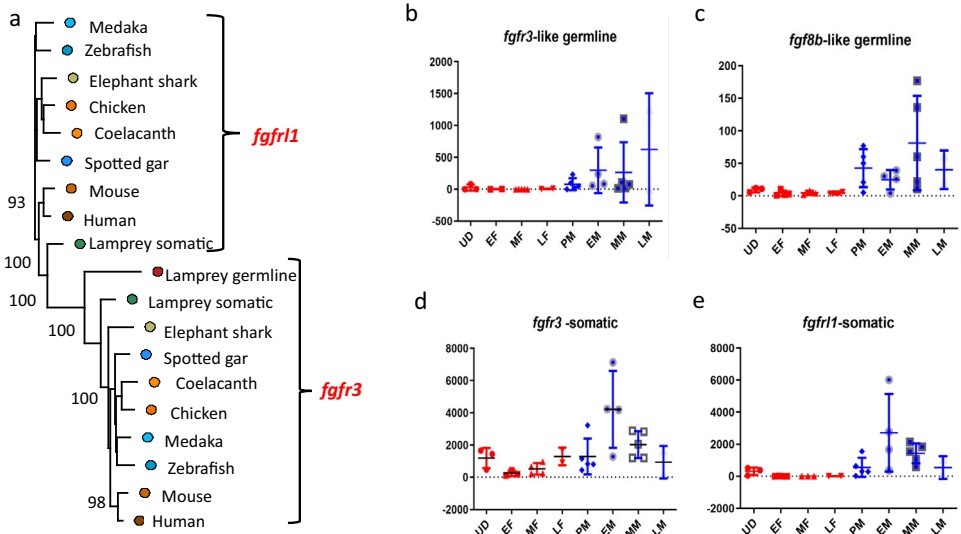

**Fig. 3 Phylogenetic relationship and expression of fibroblast growth factor (*fgf*) genes. a** Phylogeny of fgf receptors, *fgfr1* and *fgfr3*, in the sea lamprey germline and somatic genomes relative to orthologs in other vertebrates. Expression of **b** *fgfr3*-like receptor and **c** putative ligand *fgf8b* in the germline genome and **d** *fgfr3* and **e** *fgfrl1* in the somatic genome. **b**–**e** The *Y*-axis is gene counts and the *X*-axis is stage, where UD is undifferentiated larvae; EF, MF, and LF are early, mid, and late females; and PM, EM, MM,Q6 and LM are prospective, early, mid, and late males, respectively.

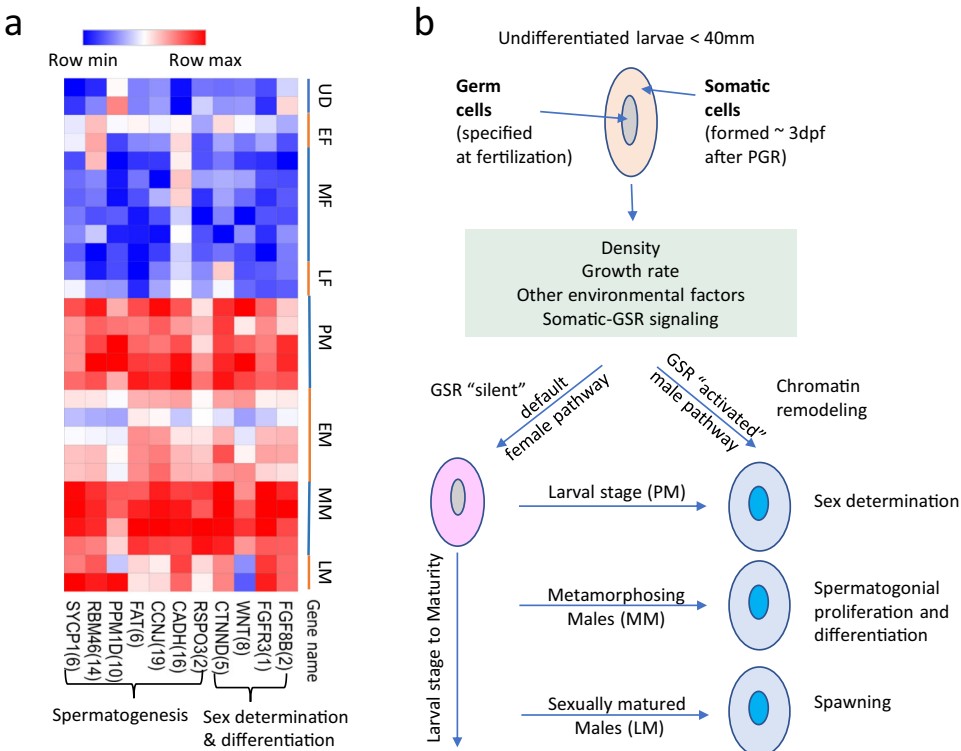

**Fig. 4 Stage-specific expression of key GSGs and a model for the role of the GSR in sea lamprey sex determination/differentiation. a** Heatmap of the median expression of paralogs in gene families present in GSR that have known roles in sex determination or spermatogenesis in other vertebrates. The numbers in brackets are the number of paralogs of these genes present in GSR. UD represents undifferentiated larvae, EF, MF and LF represent early, mid and late females respectively and PM, EM, MM and LM represents prospective, early, mid and late males, respectively. **b** A hypothesized model for the potential role of GSR in sea lamprey sex determination/differentiation pathway.

and mid-stages males, when gonadal germ cell specification and spermatogonial development are occurring, respectively (Fig. 4a). This suggests that the GSR is likely to play a role in gonadal sex determination and differentiation as well as spermatogenesis in sea lamprey.

Given these novel research findings, we propose a working model for sex determination in sea lamprey (Fig. 4b). We propose that in the undifferentiated lamprey gonad, in response to environmental cues (e.g., density, growth rate) and somatic-GSR molecular cross-talk, a decision is made to either open the

chromatin of the GSR or let it remain predominantly silenced. If the GSR remains silent, the gonad will initiate development of oocytes and continue development as a female. In contrast, if the GSR is opened, we propose that a cascade of signaling events ensue, and the gonad will commit to develop into a testis. The gonad of prospective males will remain relatively quiescent until metamorphosis, at which point the GSR will again play important roles in spermatogonial differentiation, and later in spermiation in adults, though these processes are likely controlled in concert with a host of genes expressed in the somatic genome. This model suggests a mechanism by which this early vertebrate without heteromorphic sex chromosomes may combine environmental and genetic information to determine sex.

## Conclusion

The study of PGR events and their effects on gonadal development and sex determination represent a burgeoning field in evolutionary biology. Our results suggest that the genes present in the GSR in sea lamprey are likely involved in the crucial processes of sex differentiation and testicular development, and might be involved in sex determination. We found GSGs are most highly expressed in prospective males and in males undergoing spermatogonial differentiation, while all but a few GSGs had low overall expression in females. Assuming females harbor the same GSR as males, our data suggests that the factors controlling epigenetic modification of the GSR could be pivotal for sex determination and differentiation. Further, we found that orthologs or paralogs of many of the genes identified in the GSR play known roles in gonad differentiation or sex determination in other vertebrates. However, the complete mechanism of sex determination and differentiation and what controls these processes cannot be determined using only a transcriptomics approach. Functional genomics studies are needed to address questions identified here, such as whether females harbor the same GSR as males, the role of chromatin accessibility and methylation of the GSR in gonad development, and what core genes influence the sex-determining pathway (such as *fgf8/fgfr3*). The long and complex life history of sea lamprey has previously hindered attempts to resolve the enigma of sex determination[17]. However, given recent advances in performing gene knockout experiments in lampreys[58], many of these analyses are now imminently tractable.

## Methods

**Sample preparation and RNA extraction**. Sea lamprey from different life-history stages were collected by collaborators using these samples for other projects. An Abbreviated Protocol for Minimal Animal Involvement form completed at the University of Manitoba determined that an Animal Use Protocol (AUP) was not required because live sea lamprey were not handled by us for the purposes of this project, and no animals were sacrificed or manipulated solely to provide us with tissue.

Larval sea lamprey were collected by backpack, pulsed DC electrofishing in tributaries of the Richibucto River, New Brunswick, Canada, or in tributaries of Lake Huron and Lake Michigan in the Great Lakes basin (Supplementary Table 1). Larvae were transported or shipped live to Wilfrid Laurier University, Waterloo, ON, sorted according to size, and transferred to 110 L holding tanks supplied with aerated well water at a flow rate of 1.0–2.0 L/min. The larvae were monitored for external signs of metamorphosis (e.g., changes in eye and oral disc morphology) and then euthanized at the desired stages. The brain and gills, required for other projects, were dissected and placed in RNAlater. With the remaining carcass (posterior to the last branchial pore), RNAlater was injected into the gut to perfuse the intestine, liver, gallbladder, kidneys, and gonad. The carcass with organs was then placed in a 10 mL Falcon tube and filled with RNAlater to saturate the tissues thoroughly. Dissections were completed as rapidly as possible to reduce any potential RNA degradation. Samples were kept at 4 °C for 24 h, stored at –80 °C, and then shipped to the University of Manitoba on dry ice, and stored at –80 °C upon arrival. The gonads were subsequently dissected out and placed in a 1.5 mL centrifuge tube with 1 mL RNAlater and kept at –20 °C. Sex was identified during dissection based on the physical inspection with the naked eye (i.e., the ovary is larger and has a different texture than the testis or undifferentiated gonad), and the gonadal stage was identified by a combination of visual inspection and inferences based on larval size and stage of metamorphosis[17] (Supplementary Table 1).

Adult sea lamprey were captured in traps near the mouth of the Black Mallard River or Ocqueoc River, MI, during their upstream (spawning) migration (Supplementary Table 1). Lamprey were euthanized, length and weight measurements were taken, and ~35 mg gonad was flash frozen in a 2.0 mL centrifuge tube and kept on dry ice (April 2018) or placed in a 1.5 mL centrifuge tube with 1 mL RNAlater and kept at –20 °C (June 2018). Samples were shipped to the University of Manitoba on dry ice, and stored at –80 °C. Total RNA was isolated from the gonadal tissue using the RNeasy Mini kit (Qiagen, USA) according to the manufacturer's protocol. The extracted RNA was treated with RNase-free DNase set (Qiagen, USA) to remove residual genomic DNA. RNA quantity and quality was assessed using a NanoVue Plus spectrophotometer. The RNA samples were preserved at –80 °C.

To obtain a comprehensive representation of gene expression, RNA from individuals at the same stage of development and same sex was pooled. Early males ($n = 4$) were those identified by external morphological characteristics to be in the early to mid stages of metamorphosis and thus presumed to be in the early stages of spermatogonial differentiation, that is, in the process of producing Type A spermatogonia (Supplementary Fig. 1, Supplementary Table 1)[17]. Mid males (metamorphosing stage 7 and immediately post-metamorphosis; $n = 6$) were presumed to be undergoing spermatogonial proliferation and differentiation and producing Type A and Type B spermatogonia, while late males were sexually mature ($n = 2$). In early females ($n = 2$), ovarian differentiation had been initiated and/or completed (i.e., with a number of small growing oocytes in the gonad), mid-stage females ($n = 6$) had completed oocyte differentiation and were arrested in meiotic prophase with larger growing oocytes, and late females were sexually mature ($n = 2$). In addition to samples that were definitively male and female, larvae that had histologically undifferentiated gonads and were below the size at which ovarian differentiation occurs ($n = 2$) and presumptive male larvae with histologically undifferentiated gonads but beyond the size at which ovarian differentiation is complete ($n = 4$) were included.

**Library preparation, Illumina sequencing, and data filtering**. High-quality RNA from 28 gonad samples was sent to Genome Quebec, McGill University, Montreal, to construct a cDNA library and perform RNA sequencing. Messenger RNA (mRNA) was isolated using poly-A isolation and non-normalized libraries prepared using the Illumina TruSeq DNA Kit and Epicentre Script Seq Kit. Sequencing was performed in both forward and reversed directions and 100 base pair (bp) reads were generated on an Illumina Hi-Seq 4000 PE100. The resulting RNA-Seq paired-end (PE) reads were checked for quality control using FASTQC (v0.11.8)[59], and low-quality sequences and adapters were trimmed with Trimmomatic (v0.36)[60], using ILLUMINACLIP: TruSeq3-PE-2. fa: 2:15:10 LEADING:5 TRAILING:5 SLIDINGWINDOW: 4:5 MINLEN:50 and a quality score threshold of Phred-33.

**Combining reference and *de novo* assemblies**

*Generating comprehensive gonadal superTranscriptome.* The software pipeline Necklace[19], was used to generate a merged superTranscriptome derived from three sources: 1) a genome-guided alignment using the sea lamprey reference genome, 2) a *de novo* assembly using Trinity, and 3) a reference-based proteome from other chordate species. For the genome-guided assembly, the 28 gonadal transcriptomes were mapped to the Vertebrate Genome Project (VGP) sea lamprey reference germline genome (https://ftp.ncbi.nlm.nih.gov/genomes/all/GCF/010/993/605/GCF_010993605.1_kPetMar1.pri/GCF_010993605.1_kPetMar1.pri_genomic.fna.gz) and associated gene annotation file (https://ftp.ncbi.nlm.nih.gov/genomes/all/GCF/010/993/605/GCF_010993605.1_kPetMar1.pri/GCF_010993605.1_kPetMar1.pri_genomic.gff.gz) available at NCBI. Reads were aligned to the sea lamprey genome using HISAT2, and StringTie[61] was used to assemble transcripts, some of which map to known genes and some of which are novel (MSTRG IDs). For the third tier of the Necklace pipeline, reference proteomes from a nonteleost fish, spotted gar (*Lepisosteus oculatus*) (https://ftp.ncbi.nlm.nih.gov/genomes/all/GCF/000/242/695/GCF_000242695.1_LepOcu1/GCF000242695.1_LepOcu1_protein.faa.gz), and a cartilaginous fish, elephant shark (*Callorhinchus milii*) (https://www.ncbi.nlm.nih.gov/genome/689?genome_assembly_id=49056) were used.

In the second step, a *de novo* assembly of reads was generated with Trinity[62] for all 28 samples. The assembled transcripts from genome-guided and *de novo* assembly were sorted into three groups: annotated transcripts that align to the reference genome (known genes), transcripts that align to the reference genome but are not found in the reference annotation (reference-based novel genes), and unmapped novel transcripts – those that align to the spotted gar/elephant shark proteome (*de novo*-specific genes). These three groups were merged into a single superTranscriptome and used for the second stage of the analysis: gene counting and differential expression analyses. The Necklace pipeline allows for the identification of novel transcripts yet generates a compact and comprehensive superTranscriptome, while preventing the introduction of false chimeras generated during *de novo* assembly. The step-by-step workflow of Necklace pipeline is illustrated in Supplementary Fig. 14.

In total, we identified 42,479 genes in the sea lamprey germline genome, of which 20,630 overlapped with those annotated by NCBI (representing ~94% of the total number of genes in the VGP annotation), 21,808 were identified *de novo* through StringTie, and 40 Trinity *de novo* assembled transcripts matched

sequences in the spotted gar/elephant shark reference proteome by homology. However, since the genomic location of these 40 homology-based sequences could not be ascertained, they were discarded from further analyses. Of the remaining 42,439 sequences, tRNA, rRNA, and lncRNAs (long non-coding RNAs) were removed, retaining 18,945 protein-coding transcripts (16,328 from the VGP annotation and 2,617 novel transcripts, which is ~14% of the total gene list) (Supplementary Fig. 15). Those 18,945 genes pertain to 12,583 unique gene names, which would be a lower limit on the actual number of genes identified, since paralogous genes may be assigned the same gene name.

*Gene counts.* Reads from each of the 28 gonadal transcriptomes were subsequently aligned to the merged superTranscriptome, and gene counts were extracted and filtered. These gene counts are used for further downstream analysis, i.e., in differential gene expression analysis, identifying sex-biased and sex-specific transcripts and genes.

### Functional annotation and identifying orthogroups

*Functional annotation.* All of the 18,945 putatively protein-coding genes generated from the Necklace pipeline were annotated using the Trinotate pipeline (v3.2.0)[63] following the method described at (http://trinotate.github.io/). Initially, Transdecoder (v5.5.0) was used to obtain the expected start and stop sites of protein translation from the assembled superTranscriptome. Then each transcript and protein sequence were searched against the SwissProt database using blastx and blastp. The HMMER algorithm was used to search PFAM (ran in hmmer (v3.2.1)) for protein domain identification, signalp (v4.1 f) and tmHMM (v2.0c) were used to predict the signal peptide and transmembrane regions, respectively, and Rnammer (v1.2) was used to identify rRNA transcripts which were automatically removed in a later stage of the pipeline. In the final stage, the results from blast searches were combined with the other functional annotation data and loaded into the Trinotate.SQLite database: an e-value of 1e-5 was used as the threshold to generate the functional annotation report.

*Orthogroup identification.* The homology between the genes in our annotated sea lamprey gonadal superTranscriptome was compared to genes in 11 chordate species chosen to represent important time points in chordate evolution using the Ortho-Finder pipeline. Protein sequences were obtained from human (ftp.ensembl.org/pub/release-103/fasta/homo_sapiens/pep/Homo_sapiens.GRCh38.pep.all.fa.gz), mouse (*Mus musculus*) (ftp://ftp.ensembl.org/pub/release-102/fasta/mus_musculus/pep/Mus_musculus.GRCm38.pep.all.fa.gz), zebrafish (ftp.ensembl.org/pub/release-103/fasta/danio_rerio/pep/Danio_rerio.GRCz11.pep.all.fa.gz), chicken (*Gallus gallus*) (ftp.ensembl.org/pub/release-103/fasta/gallus_gallus/pep/Gallus_gallus.GRCg6a.pep.all.fa.gz), medaka (*Oryzias sinensis*) (ftp.ensembl.org/pub/release-103/fasta/oryzias_sinensis/pep/Oryzias_sinensis.ASM858656v1.pep.all.fa.gz), spotted gar (ftp.ensembl.org/pub/release-103/fasta/lepisosteus_oculatus/pep/Lepisosteus_oculatus.LepOcu1.pep.all.fa.gz), elephant shark (ftp.ensembl.org/pub/release-103/fasta/callorhinchus_milii/pep/Callorhinchus_milii.Callorhinchus_milii-6.1.3.pep.all.fa.gz), coelacanths (ftp.ensembl.org/pub/release-103/fasta/latimeria_chalumnae/pep/Latimeria_chalumnae.LatCha1.pep.all.fa.gz/), hagfish (*Eptatretus burgeri*) (ftp.ensembl.org/pub/release-103/fasta/eptatretus_burgeri/pep/Eptatretus_burgeri.Eburgeri_3.2.pep.all.fa.gz), and amphioxus (*Branchiostoma belcheri*) (https://ftp.ncbi.nlm.nih.gov/genomes/all/GCF/001/625/305/GCF_001625305.1_Haploidv18h27/GCF_001625305.1_Haploidv18h27_protein.faa.gz). OrthoFinder uses the complete list of known protein sequences from all included taxa to find putative orthologs, and then creates orthogroups with related sets of orthologs. OrthoFinder exports multiple sequence alignments and rooted gene trees for all orthogroups, which can be used to infer gene duplication events. Overall, in this study, 93.1% of the genes in the 12 chordate species were assigned to one of 27,364 orthogroups, and 5606 orthogroups contained representatives of all 12 species.

### Prediction of germline-specific region (GSR) and genes by enrichment analysis

*Identifying GSGs in the GSR.* Although the GSR in sea lamprey has been identified for a previous germline assembly (gPmar100)[20], when the new VGP germline genome was deposited on NCBI, the corresponding positions were not available. Following the protocol from[20], germline enrichment was calculated using the DifCover program by calculating differences in read depth between a single germline sample (sperm) and a single somatic sample (blood) from the same male. We downloaded and mapped the same sperm (SRR5535435) and blood (SRR5535434) samples they had used from their previous analysis to identify the GSR coordinates in the newly deposited VGP genome in order to facilitate our downstream transcriptomic analyses. The DNAcopy output file was generated by following step by step workflow with default settings described in the DifCover pipeline (https://github.com/timnat/DifCover) (see Supplementary Fig. 14)[64]. This DNAcopy output file was then used to identify GSR from the new chromosome level assembly, VGP germline genome, and the DNAcopy output file. Later, the GSGs were identified by extracting all genes that fell within regions having an enrichment score >2 using bedtools (v2.29.0) with the aid of the genome-based annotation file (generated in Necklace pipeline).

Initially, we identified 1845 GSGs by extracting the DNAcopy output file from VGP genome; however, only 783 protein-coding GSGs were retained with gene counts after the initial filtration steps discussed in previous section. In the next step, we sorted genes based on their location: if two genes with the same name had overlapping start and end points, the canonical transcript was retained, which reduced the number of genes to 672. In the final step, we extracted the protein sequences associated with each of these genes from the transdecoder pep file and removed ambiguous sequences. The final list consisting of 638 GSGs was merged with the Trinotate annotation report to assign putative gene names for the novel genes, and with the reference annotation for genes identified by VGP. In total, 163 unique gene names were assigned to the 638 GSGs, 70 of them in a single copy, and the remaining 93 in 2–77 copies per gene family (Supplementary Fig. 2).

Given our finding that genes in the GSR are highly expressed during gonad development (see Results and Discussion), we wanted to assess whether all or a subset of the genes in the GSR are also expressed during early embryonic development. To this end, we downloaded paired-end RNA-seq read data for embryos sampled at 1 dpf (SRR3002837), 2 dpf (SRR3002840), 2.5 dpf (SRR3002843), 3 dpf (SRR3002846), 4 dpf (SRR3002849), and 5 dpf (SRR3002852) from the SRA database (https://www.ncbi.nlm.nih.gov/sra?linkname=bioproject_sra_all&from_uid=306044). Reads were aligned to the VGP genome using HISAT2 (v 2.2.1)[61], and assembled into transcripts, and gene and transcripts counts were obtained per sample using Stringtie (v 2.0)[61]. In the next step, we extracted the embryo-expressed GSGs using the merged annotation file generated in the previous step and the DNACopy output file generated from DifCover analysis (Supplementary Fig. 14). We considered only genes available in the reference annotation that mapped to our GSR, which resulted in 184 GSGs (after filtering ncRNA (non-coding RNAs), rRNA (ribosomal RNAs), and pseudogenes from the reference annotation) expressed in early embryonic development of which 149 genes overlapped with those expressed in our gonad samples. The gene count file was used to compare gene expression in the GSR pre-PGR (1dpf, 2 dpf, and 2.5 dpf) and post-PGR (3 dpf, 4 dpf, and 5 dpf) and between gonads and embryos (both pre- PGR and post-PGR).

*Identifying somatic copies of GSGs.* To identify putative paralogs of GSGs in the somatic genome, the list of all genes was sorted based on the unique gene names obtained from either the Trinotate annotation report or from the reference annotation. Of the 163 unique genes identified in the GSR, 89 were found to have either a single or multiple putative paralogs in the somatic genome, of which 31 were matched to a unique OrthoGroup by OrthoFinder (Supplementary Fig. 16).

*Comparison of male-biased gene expression across stages and genomes.* The genome-wide raw gene counts were converted to normalized counts using DESEQ2[65], and the log2 expression of genes compared in a sex- and stage-specific manner for both genomes. To assess global differences in sex-biased gene expression, we compared the density of the relative log2(male:female normalised expression) of all genes in the somatic genome vs GSR. To compare the difference in expression between GSGs against their somatic paralogs by both sex and stage, we calculated the mean normalized log2 gene count and visualised the data with a heat map.

To assess whether GSGs exhibit stage-specific sex-biased expression, we first generated a list of genes exhibiting strong male-biased expression. Traditional differential gene expression analyses use a threshold log-fold change between conditions to identify DEGs, but for this analysis, we wanted to identify genes with low or no expression in females but high expression in males. Thus, first a differential expression analysis of all genes was performed using DESeq2[65] and EdgeR[66], and the union of all genes exhibiting a log-fold change (logFC) >2 in males:females from both pipelines was extracted. This identified 6088 male-biased genes; this list was again filtered to include only genes with a total gene count >1000 in males, and the top 20% of these genes were considered to be strongly male-biased (total 1270 genes, 409 in the GSR, 861 in the somatic genome) (Supplementary Fig. 17). We then compared the relative expression of each gene by stage and genome using a repeated measures mixed model in which the proportion of genes expressed in that stage was the response and the model was gene(genome) + stage + genome*stage, with gene as a random effect, and stage as the repeated measure.

*Functional enrichment analysis.* The list of genes identified in the GSRs was submitted for pathway analysis using the human protein-coding genes as background using PANTHER (v14) (http://pantherdb.org)[67]. The PANTHER GO-slim molecular process terms associated with each gene were used for an over-representation test[67], in which the Fisher exact test was performed to assess the significance of terms at an FDR of 0.05. Additionally, we used the gene ontology (GO) terms associated with the GSGs identified by Trinotate[7], and visualized them in REVIGO (http://revigo.irb.hr/)[68] in a scatter plot that shows cluster representatives in a two-dimensional space derived by GO terms with semantic similarity measure and clustering set at 0.9 overall. Terms were plotted with size proportional to fold-enrichment above expected and color according to the log10 of the FDR P-value (Fig. 1b; see Results and Discussion).

**Phylogenetic analysis.** Given that genes in the GSRs may have unique evolutionary histories, the phylogenetic relationships of a subset of the genes in the GSRs

and their somatic orthologs were reconstructed along with orthologous/paralogous genes identified from the 11 taxa included in the OrthoFinder output. Phylogenetic trees were obtained from OrthoFinder which uses RaxML reconstruction[69]. Trees were not available for all GSGs, including the sea lamprey putative ortholog of *fgfr3*, which has been shown to be important for sex determination and differentiation in other taxa[49,50,70]. Thus, we obtained sequences for *fgfr3* for the same 11 species employed in the OrthoFinder analyses, and then performed an alignment in Maaft (https://mafft.cbrc.jp/alignment/server/) followed by ML reconstruction with RAXML. We hypothesized that genes in the GSR are under relaxed evolutionary constraint and relaxed dosage sensitivity and thus may exhibit accelerated rates of sequence evolution. However, we were unable to employ tests of dN/dS due to the difficulty of obtaining sufficiently un-gapped alignments of the coding sequence of the sea lamprey genes relative to those from jawed vertebrates. Nevertheless, phylogenetic trees were generated to understand the relationship of paralogous copies of the GSGs in the GSR to those in the somatic genome, as well as the relationship of the protein-coding sequences in sea lamprey to those in other chordate taxa.

**Reporting Summary**. Further information on research design is available in the Nature Research Reporting Summary linked to this article.

## Data availability

The RNA-sequencing reads used for this study have been deposited in the NCBI repository under the BioProject accession number PRJNA749754.

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

## Acknowledgements

We thank Dr. John B. Hume (Michigan State University), Dr. Nicholas Johnson (U.S. Geological Survey), Dr. Michael Wilkie (Wilfrid Laurier University), and Joshua Sutherby (University of Manitoba) for providing us with the samples used in this study. We also thank Arfa Khan (University of Manitoba) for her guidance during sample processing and RNA extraction. This research was funded by the Natural Sciences and Engineering Research Council of Canada (NSERC) Discovery Grants program (MFD, SVG), the Great Lakes Fishery Commission Sea Lamprey Research Program (MFD), and the University of Manitoba Graduate Enhancement of Tri-Council Stipends program (MFD).

## Author contributions

T.Y. – Performed wet laboratory work, helped design and perform data analysis, wrote first draft of paper, helped edit all revised versions. P.G. – provided assistance for the identification of the GSR and helped edit the paper at all stages. M.D. – obtained funding, coordinated sample collection and edited the paper at all stages. S.V.G. guided and performed data analysis, helped draft the first and edited the final version of the paper.

## Competing interests

The authors declare no competing interests.

## Additional information

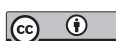

