## [Peer Review File · Communications Biology]

Reviewers' comments:

Reviewer #1 (Remarks to the Author):

As in many other organisms, programmed genome rearrangement (PGR) is known to occur in the lamprey, in which approximately 20% of the genome is removed during early development from somatic cells, and only germline cells retain the full genome complement. The germline genome contains ~500 Mb that is eliminated during PGR. Yasmin and colleagues used a transcriptomics approach to identify 638 germline-specific genes, many of which mapped to a germline-specific region of the genome. Interestingly, they found that many of these genes are highly expressed during spermatogenesis, but not during oogenesis, even though they appear to be present in undifferentiated larvae and females. Male-biased expression of germline-specific genes, largely confined to the differentiating male gonad, led Yasmin and colleagues to suggest GSGs likely play a role in male-specific sex determination and differentiation.

The strength of this paper is the transcriptomics approach to identification of genes in the germline-specific region of the genome that are jettisoned from the somatic lineage. This is a very powerful approach that will advance our knowledge of sex determination and differentiation in a vertebrate for which little is known regarding these mechanisms, furthering understanding of the evolution sex determination and differentiation mechanisms. The authors do a nice job of demonstrating germline expression of GSG genes, compared to their somatic paralogues, and biased toward male differentiation. Interestingly, many of the genes identified in the GSR were mapped to a single chromosome (Chromosome 81) with others in unassembled scaffolds, supporting the view that PGR in lamprey likely involves chromosome elimination. Members of this team have been instrumental in demonstrating there may be environmental cues that are involved in sex determination in lamprey, particularly related to larval density, but this work is the first to demonstrate at a genomic level that sex differentiation in lamprey involves many of the same key genes known to play roles in sex differentiation in other organisms.

The primary weakness of the paper is the interpretation of their data that "...gene expression in the sea lamprey GSR may function to control sex determination..." The authors have demonstrated very clearly that expression of germline-specific genes correlates strongly with male-specific sex differentiation. However, they have not established these genes control the mechanism of sex determination (correlation is not causation). This would require a functional genomic approach to first identify a causal link between the upstream, possibly environmental cue(s), with downstream mechanisms (e.g., fibroblast growth factor activity) that may drive the sex determination decision. The authors address this in the paragraph beginning at line 371, where they propose that in response to environmental cues, "a decision is made" resulting in activation of the GSR. However, what controls this decision cannot be determined using a transcriptomics-only approach.

The authors note the lack of conserved synteny among genes in the GSR with other chordates. This is noted in the Abstract, and again in the concluding paragraph. However, I did not find any expansion of this idea among the Results and Discussion section of the paper. This idea could be strengthened with some elaboration. Since numerous genes are identified that play conserved roles in sex differentiation, it would be both interesting and important to elaborate on the conservation of genes involved in a conserved process (sex differentiation) coupled with the lack of synteny.

Minor:

Line 35: Please change agnathan to agnathans.

Line 253: Please correct misspelling of "prior"

Reviewer #2 (Remarks to the Author):

In this manuscript, the authors investigated the RNA expression from gonads at various stages of male and female sea lamprey and identified ~600 germline-specific genes (GSG) that reside in germline-specific regions (GSR) that exist in the sperm DNA but not in the somatic genome. They

found that ~400 of GSGs are expressed in developing testis. Their comparison of transcripts from different testicular developmental stages further showed that a higher proportion of GSGs are expressed in prospective males and in mid-testicular development.

On the other hand they found that most GSGs expressed very lowly in undifferentiated gonads and female gonad. They also found that almost all of the genes in the GSR show male-biased expression while genes in the somatic genome tend to be expressed in the both sexes. Moreover, they showed that among the 1270 genes showing male-biased expression, 409 were in the GSR and 861 were in the somatic genome, indicating that the GSR are enriched in male-biased genes.

They further found that GSGs include genes that are known to be involved in sex differentiation and/or determination in other vertebrates. Therefore, together with the above observations, the authors suggested that GSGs play important roles in spermatogenesis and maybe also in sex determination.

They also found that some of the transcripts from female gonads were derived from the GSR and thus claimed that the GSR retain in the female germline. Therefore, the authors suggested that the male-biased expression of GSGs is not regulated by sex-biased programmed genome rearrangement (PGR) but is likely regulated epigenetically by DNA methylation.

I found the authors analyzed the RNA-seq data efficiently to extract many interesting biological implications about PGR and sex development in sea lamprey. I recommend this manuscript to be published with minor revision. I just have a few comments that may make the manuscript even better.

Comments:

1) This study does not include any functional investigation of GSGs. So, I think "plays a key role in spermatogenesis" in the title is an overstatement.

2) Are GSGs only genes in GSR? If not, how many genes are predicted in GSR and how many of them show male-biased expression patterns?

3) Investigation of GO term enrichment may also help us to understand in which biological pathways GSGs are involved.

4) It would be nice to give some explanation/speculation for why many GSGs are expressed in PM and MM but are down regulated in EM.

5) While the authors stated that "410 transcripts from female gonad samples that mapped to either known or novel exons in the GSR" (line 163), it is unclear for me how much of GSR is covered by these 410 transcripts. Are they cover a large fraction of GSR or only very small portion of them? Also, it is unclear for me if these transcripts were all come from female germline cells or if some of them derived from undifferentiated germ cells in gonads whose sexes were not yet determined. From these uncertainties, I am not sure if their observation supports presence of the same GSR in male and female germline cells.

6) I think the authors suggestion that GSGs are regulated epigenetically by DNA methylation is very reasonable. However, it is then unclear for me why they have to be in GSR and what is the role of PGR in gonadal sex development/differentiation. It would be nice to give some more discussion for these points.

Reviewer #1:**Comment:**

The primary weakness of the paper is the interpretation of their data that "...gene expression in the sea lamprey GSR may function to control sex determination..." The authors have demonstrated very clearly that expression of germline-specific genes correlates strongly with male-specific sex differentiation. However, they have not established these genes control the mechanism of sex determination (correlation is not causation). This would require a functional genomic approach to first identify a causal link between the upstream, possibly environmental cue(s), with downstream mechanisms (e.g., fibroblast growth factor activity) that may drive the sex determination decision. The authors address this in the paragraph beginning at line 371, where they propose that in response to environmental cues, "a decision is made" resulting in activation of the GSR. However, what controls this decision cannot be determined using a transcriptomics-only approach.

The authors note the lack of conserved synteny among genes in the GSR with other chordates. This is noted in the Abstract, and again in the concluding paragraph. However, I did not find any expansion of this idea among the Results and Discussion section of the paper. This idea could be strengthened with some elaboration. Since numerous genes are identified that play conserved roles in sex differentiation, it would be both interesting and important to elaborate on the conservation of genes involved in a conserved process (sex differentiation) coupled with the lack of synteny.

Response: We have edited the manuscript throughout to soften the conclusion that the GSR plays a role in sex determination *per se*, and clarified this in the conclusions. A transcriptomics-

only approach will not be able to determine the whole mechanism of sex determination and differentiation, as well as what governs these processes. Our findings point to a substantial link between gonadal development and male-biased germline-specific expression. This association suggests that these genes are involved in spermatogenesis and/or sex differentiation. Similarly, the activation of this male-specific gene set at a certain embryonic stage is linked to gonadal differentiation, which is an important feature of sex determination. Our findings show that the germline-specific region may play a role in determining sex. Future work can build upon the strong correlations that we have identified using the functional genomics work that would be required to move from a "may" to a "does" or "does not". We also included a specific statement in the conclusion (Line 396–399) saying that the complete mechanism of sex determination and differentiation and what controls these processes cannot be determined using only a transcriptomics approach. Functional genomics studies are needed to address questions identified here...”

Minor:

Line 35: Please change agnathan to agnathans.

Response: Corrected

Line 253: Please correct misspelling of “prior”

Response: Corrected

Reviewer #2:

Comments:

1) This study does not include any functional investigation of GSGs. So, I think “plays a key role in spermatogenesis” in the title is an overstatement.

Response: We decided to make the title broader and less definitive. The new title of the paper is “Pervasive male-biased expression throughout the germline-specific regions of the sea lamprey genome supports key roles in sex differentiation and spermatogenesis”

2) Are GSGs only genes in the GSR? If not, how many genes are predicted in GSR and how many of them show male-biased expression patterns?

Response: Firstly, we classify all genes within the GSR as GSGs and have now clarified this in the text on line 68: “We identified 638 genes in the GSR, which we will refer to as germline-specific genes (GSGs). Many of these GSGs existed as multiple germline-specific paralogs, such that the 638 GSGs belonged to 163 gene families.”

Regarding male-biased gene expression - Thank you for this comment, as we have now clarified what we mean by male-biased genes throughout the paper. Of the 638 genes in the GSR, all are expressed in the male gonad, but not all are expressed in differentiated female gonad or in undifferentiated larvae. We now give the basic statistics about the expression of genes in the GSR in the paragraph on lines 125-138. Although all 638 genes are expressed in male, 31 exhibit low median expression (a median count <5 across samples), while the median count of genes expressed in the GSR in female samples is ~10 and 445 genes have <5 counts. In subsequent analysis in this section we calculated the log₂FC male:female gene expression across both the somatic and GSR genomes and present this data in a heat map (1a), density graph (1b) and

chromosome wide expression (1c). This shows that genes in the GSR do exhibit overall male-biased expression, but those in the somatic genome do not (see esp. Figure 1b).

Additionally, in a second analysis, we examine whether male-biased genes in the GSR are enriched for expression in one stage of male (and prospective male) development relative to male-biased genes in the somatic genome. To do this, we used a stricter threshold to define male-biased genes (described in the methods on lines 600-617), in which we selected the top 20% of male-biased genes (see Supplementary figure 16, most of these genes have \log_2 FC in males >10). Using a repeated measures ANOVA we tested whether these strongly male-biased genes have higher expression in one stage of male development compared to highly male-biased genes in the somatic genome. This revealed that strongly male biased genes are enriched in prospective males and in males in mid-developmental stages.

3) Investigation of GO term enrichment may also help us to understand in which biological pathways GSGs are involved.

Response: Our initial submission did contain results from a GO term enrichment analysis that were present in Fig 1b and supported enrichment of terms including “reproductive system development”. This figure is still included in the main text, and we have now clarified some of the results in lines 150-162. We also provided supplemental tables (Supplementary Table 5 and 6) and Supplementary Fig 6 to provide further evidence of our findings.

4) It would be nice to give some explanation/speculation for why many GSGs are expressed in PM and MM but are down regulated in EM.

Response: We provide our rationale for there is enrichment of GSGs in PM and MM on lines Line 237-246 (see below):

Here, we find evidence that the GSGs show comparatively higher expression in both presumptive males and males in mid-testicular development (Supplementary Fig. 1). In mid-developmental males, the germ cells are undergoing mitotic proliferation and producing spermatogonial Type B cells (see Supplementary Figure 1). In prospective males, the gonads are histologically undifferentiated (and may remain so for another 1–3 years¹⁷), while females from the same size class have oocytes arrested in Meiosis I. The finding of high gene expression of GSGs in undifferentiated prospective males but not in females of the same size class suggests that the GSR plays an important role in sex differentiation and potentially sex determination in sea lamprey, with high expression of GSGs leading to testicular differentiation and development in males and gene silencing resulting in ovarian differentiation in females.

5) While the authors stated that “410 transcripts from female gonad samples that mapped to either known or novel exons in the GSR” (line 163), it is unclear for me how much of GSR is covered by these 410 transcripts. Are they cover a large fraction of GSR or only very small portion of them? Also, it is unclear for me if these transcripts were all come from female germline cells or if some of them derived from undifferentiated germ cells in gonads whose sexes were not yet determined. From these uncertainties, I am not sure if their observation supports presence of the same GSR in male and female germline cells.

Response:

Thank you – this is an important point, that we have now clarified.

A possible explanation for this observation could be that females do not have the same GSR as males, since the reference genome for sea lamprey was generated using sperm DNA. However, this does not appear to be the case: genes expressed in the developing female gonad were found on chromosome 81 as well as many of the unassembled scaffolds (Supplementary Table 3). To examine this more closely since RNA-seq reads may map to spurious locations, we aligned individual BAM files from both male and female gonad samples to the indexed superTranscriptome and annotation files using the Integrative Genome Viewer (IGV)²¹. This revealed that for the few genes expressed in both testes and ovaries, the reads aligned to overlapping genomic (exons) structures (e.g., see Supplementary Fig. 8a, 8b). This suggests that females harbour the same GSR as males but that it may be silenced via epigenetic controls such as DNA hypermethylation or histone modifications.

6) I think the authors suggestion that GSGs are regulated epigenetically by DNA methylation is very reasonable. However, it is then unclear for me why they have to be in GSR and what is the role of PGR in gonadal sex development/differentiation. It would be nice to give some more discussion for these points.

Response: We have now edited the sections of the manuscript discussing this and the role of epigenetics in sex differentiation and determination. The majority of investigations into the function of PGR in vertebrates has found that it is associated with the elimination of sex chromosomes, and thus it is argued that PGR can be an extension of dosage compensation in which epigenetic inactivation of genes on sex chromosomes is used to equalize gene expression in males and females.

Most organisms that undergo PGR, such as ciliates, sciarid flies, songbirds, hagfishes and lampreys, exhibit DNA and histone methylation prior to the elimination of PGR-targeted

region and these epigenetic changes may contribute to gene silencing. On the other hand, proteins involved in the deposition of histone modifications or the opening of dense chromatin are known to play a key role in germ cell fate determination and gonad development in other species. So, it is possible that comparable epigenetic silencing via DNA and histone methylation that plays a role in PGR could also contribute to regulation of the GSR during gonad development. Given the strong association of the GSG's with male gonad development, particularly in prospective males that have undifferentiated gonads, the most parsimonious hypothesis is that regulation of the GSR in combination with the expression of a few key genes (e.g., *fgf8/fgfr3*) (some of which may be in the somatic genome) are responsible for initiating male gonad development.

REVIEWERS' COMMENTS:

Reviewer #2 (Remarks to the Author):

I found that the authors have responded properly to my previous comments. I just have one remaining concern about GSR in female germline:

While the authors are probably right about "genes expressed in the developing female gonad were found on chromosome 81 as well as many of the unassembled scaffolds (Supplementary Table 3)", it is very difficult to extract such information from Supplementary Table 3. Is it possible to analyze the gene expression in female gonad like as Fig 1c but just showing the female (but not the male/female ratio)?

Response to reviewer #2

Reviewer #2 (Remarks to the Author):

I found that the authors have responded properly to my previous comments. I just have one remaining concern about GSR in female germline:

While the authors are probably right about "genes expressed in the developing female gonad were found on chromosome 81 as well as many of the unassembled scaffolds (Supplementary Table 3)", it is very difficult to extract such information from Supplementary Table 3. Is it possible to analyze the gene expression in female gonad like as Fig 1c but just showing the female (but not the male/female ratio)?

>> We have added the additional figure into the supplementary Figures. It is now Supplementary Figure 7, cited on line 224.